# INFERENCE-BASED REWARDS FOR REINFORCEMENT LEARNING

## ABSTRACT

A central challenge in reinforcement learning (RL) is defining reward signals that reliably capture human values and intentions. Recent advances in vision–language models (VLMs) suggest they can serve as a powerful source of semantic rewards, offering a flexible alternative to environment-defined objectives. Unlike hand-crafted signals, VLM-based feedback can reflect high-level human goals such as safety, efficiency, and comfort. We first analyze the conditions under which VLM-based rewards enable effective learning. In particular, we highlight the importance of monotonicity with respect to true task performance and the satisfaction of the Markov property. When these conditions hold, VLMs provide a viable basis for reward inference. On the algorithmic side, we identify what learning strategies are best suited for such rewards. Trajectory-based methods such as policy gradient (e.g., REINFORCE, PPO) are naturally aligned with inferred returns, whereas Q-learning style algorithms are more fragile because they operate on step-wise Bellman updates (e.g., DQN) and implicitly assume the Markov property of rewards. This perspective reframes RL around reward inference rather than reward specification, highlighting both the promise of VLM-based alignment and the theoretical and practical boundaries of when such methods are effective. Experiments across control domains provide supporting evidence for these insights. In particular, monotonicity appears to align with learning outcomes, REINFORCE and PPO show greater robustness than DQN when trained with inferred rewards, and natural language prompts can guide the emergence of instruction-driven behaviors.

## 1 INTRODUCTION

Reinforcement learning (RL) has delivered striking successes in games, robotics, and control, but its practical use continues to be constrained by a fundamental problem: reward design. Standard formulations assume that the designer can specify a reward function that faithfully encodes the intended task. In practice, this assumption rarely holds. Many desirable goals, such as safety, efficiency, or comfort, are difficult to quantify numerically. Even small mismatches between the intended objective and the coded reward can lead to unsafe or unintended behavior (Rusu et al., 2017; Levine et al., 2018; Pinto & Gupta, 2016). The reward specification problem remains one of the main barriers to scaling RL to real-world settings (He et al., 2024; Anderson et al., 2021; Kadian et al., 2020; Chang et al., 2025; Hsu et al., 2023).

Recent advances in vision–language models (VLMs) suggest a promising alternative. Models such as CLIP (Radford et al., 2021) embed images and text into a shared semantic space, enabling natural language descriptions to act as reward signals. With the right prompt, an RL agent can be guided toward behaviors without requiring manually engineered objectives. Unlike hand-crafted signals, VLM-based feedback can be flexibly updated by changing the prompt and can capture high-level semantic goals. Early results have demonstrated that such inferred rewards can support RL in a variety of domains (Rocamonde et al., 2024; Baumli et al., 2023; Wang et al., 2024; Chan et al., 2023; Du et al., 2023; Sontakke et al., 2023; Yang, 2024; Yu et al., 2023; Schoepp et al., 2025; Ye et al., 2025), raising the possibility of scalable and language-driven alignment.

Yet, simply substituting environment-defined signals with VLM-based rewards does not guarantee success. Inferred rewards are noisy and context-dependent, and their usefulness depends on structural properties. We identify order-preserving monotonicity as an important condition: inferred rewards

should preserve the ordering of trajectories with respect to true performance. To support this claim, we introduce evaluation metrics such as pairwise agreement and rank correlation (Kendall's $\tau$, Spearman's $\rho$) that allow monotonicity to be measured in benchmark settings where the true reward is available. These metrics are not intended as standalone tools, but as a methodology for validating when inferred rewards provide a reliable learning signal and for guiding prompt design to maximize alignment with intended goals.

Our analysis further suggests that algorithmic families differ in their suitability for inferred rewards. Policy gradient methods align naturally with trajectory-level signals, while value-based approaches such as Q-learning rely on Bellman updates that assume step-wise Markovian rewards. When this assumption is violated, value-based methods may propagate errors, offering a possible explanation for instabilities observed in prior work with semantic reward models (Rocamonde et al., 2024; Baumli et al., 2023). We additionally highlight the role of quasi-Markov structure induced by short temporal windows of observations, which affects value-based methods more strongly than trajectory-based ones. This perspective not only helps interpret past results but also opens new directions: improving policy gradient methods to exploit trajectory-level structure more effectively, and rethinking value-based updates to accommodate non-Markovian settings.

Empirically, we validate these insights across classic control and continuous control domains. Our experiments show that trajectory-based methods such as REINFORCE and PPO can successfully leverage inferred rewards to achieve performance comparable to ground-truth rewards, while Q-learning methods such as DQN struggle when monotonicity and Markov assumptions do not hold. We further demonstrate that prompt choice directly affects monotonicity, highlighting the potential for systematic prompt design to improve alignment. We further study how prompt choice and temporal window size affect monotonicity and quasi-Markov structure, providing practical levers for improving reward alignment. Together, these results provide both practical guidance and new challenges for integrating language-driven rewards into reinforcement learning.

Our contributions are as follows:

- Conceptual framework and theoretical analysis establishes conditions under which inferred rewards support effective learning by emphasizing order-preserving monotonicity of trajectory returns as a key property.

- Algorithmic analysis examines how different reinforcement learning families interact with inferred rewards and shows that trajectory-based methods such as REINFORCE and PPO are more robust than step-wise value-based methods such as DQN.

- Practical monotonicity measure proposes metrics for quantifying monotonicity that validate inferred rewards in benchmark settings and can guide prompt design.

- Empirical results demonstrate that inferred rewards can match ground-truth performance in standard tasks, exhibit predictable behavior as monotonicity and quasi-Markov structure vary, enable novel instructed behaviors, and reveal limitations in handling semantically complex instructions.

## 2 BACKGROUND AND RELATED WORK

**Reinforcement Learning and Reward Design.** Reinforcement learning (RL) is commonly formulated under the Markov Decision Process (MDP) framework, where an agent interacts with an environment defined by the tuple $(\mathcal{S}, \mathcal{A}, P, r, \gamma)$. At each timestep $t$, the agent observes a state $s_t \in \mathcal{S}$, takes an action $a_t \in \mathcal{A}$, and receives a *scalar reward* $r_t = r(s_t, a_t)$ from the environment. The agent then learns a policy $\pi(a_t|s_t)$ that maximizes the expected discounted return. While this formulation has yielded many breakthroughs, it assumes that the environment can provide a meaningful and consistent reward signal. In practice, this assumption rarely holds. Reward functions are often hand crafted, encoding brittle heuristics or proxy objectives that agents can exploit in unexpected ways. As a result, poorly designed rewards may encourage reward hacking or other unintended behaviors. In real-world deployments such as robotics or interactive systems, the challenge is even more acute: reward signals may be unavailable, delayed, or unobservable (He et al., 2024; Anderson et al., 2021; Kadian et al., 2020; Chang et al., 2025; Hsu et al., 2023; Rusu et al., 2017; Peng et al., 2018; James et al., 2017; Hsu et al., 2023; Levine et al., 2018; Pinto & Gupta, 2016).

**Approaches Beyond Hand-Crafted Rewards.** Recognizing this challenge, a variety of approaches have been proposed. Inverse reinforcement learning (IRL) (Ng & Russell, 2000) infers reward functions from expert demonstrations or preference data, providing a principled route toward human-aligned objectives. However, IRL typically requires extensive curated data and repeated human input, limiting its scalability. Preference-based RL and RLHF further leverage human comparisons or language feedback to train reward models, but similarly do not characterize when the inferred rewards are suitable for downstream RL optimization (Christiano et al., 2017; Ibarz et al., 2018; Stiennon et al., 2020). Intrinsic motivation methods such as curiosity driven exploration (Pathak et al., 2017; Burda et al., 2018) generate task-agnostic signals that encourage agents to seek novelty, but by design they remain disconnected from external goals. Similarly, goal-conditioned reinforcement learning (Andrychowicz et al., 2017) incorporates goal vectors into policies and has shown impressive adaptability, yet it still depends on external rewards to ground those goals. Each of these directions makes important progress, yet the fundamental issue of specifying rewards that capture high-level objectives remains unresolved.

**Vision-Language Models and Supervised Approaches.** In parallel, large-scale supervised learning has advanced rapidly. Vision-language-action (VLA) models (Zitkovich et al., 2023; Brohan et al., 2022; Kim et al., 2024; O'Neill et al., 2024) leverage paired image, text, and action data to train policies capable of following natural language instructions. These models illustrate the power of multimodal training and achieve strong generalization across tasks. Yet they also highlight a limitation: because they rely on large-scale labeled data and lack the trial-and-error dynamics of reinforcement learning, they do not inherit the adaptability of reward-driven methods. Their strengths therefore lie in supervised imitation rather than in autonomous discovery.

Recent developments in pretrained vision-language models (VLMs) such as CLIP (Radford et al., 2021), VideoCLIP (Xu et al., 2021), Flamingo (Alayrac et al., 2022), GPT-4 (Achiam et al., 2023), LLAVA (Liu et al., 2023; 2024), and Qwen (Yang et al., 2025) broaden the possibilities further. These models encode rich semantics across vision and language, showing remarkable generalization to novel inputs. They have been used effectively for perception, retrieval, and auxiliary scoring. Their capacity to ground instructions in visual observations naturally suggests a role as reward providers, bridging reinforcement learning with high-level semantic goals.

**Reward Inference with VLMs.** Several works have already begun exploring this potential. Rocamonde et al. (Rocamonde et al., 2024) demonstrate that VLMs can act as zero-shot reward models, allowing agents to learn from prompts without manually engineered signals. Baumli et al. (Baumli et al., 2023) study how model capacity affects the fidelity of such rewards, and (Wang et al., 2024) integrate VLM feedback into reinforcement learning pipelines to generate task-specific signals. Other efforts such as RoboCLIP (Sontakke et al., 2023) and RL-VLM-F (Yang, 2024) analyze practical stability issues of VLM-derived rewards, though they focus on reward construction rather than determining when such rewards are theoretically or algorithmically suitable for RL. These contributions establish the feasibility of VLM-based rewards and provide valuable insight into their strengths and limitations. At the same time, most of these works treat reward models as external scorers, leaving open the question of what conditions make them effective learning signals and which algorithms can best exploit them.

**Reward-Free Frameworks.** A complementary body of research has questioned the very necessity of external rewards. Reward-free frameworks argue that agents can develop competence through other principles, such as skill discovery or representation learning. For example, DIAYN (Eysenbach et al., 2019) encourages agents to acquire diverse behaviors by maximizing discriminability, while curiosity-driven exploration (Burda et al., 2018) motivates visiting novel states. These directions elegantly bypass the brittleness of hand-designed rewards, though they stop short of offering explicit alignment with semantic or task-specific objectives.

**Position of This Work.** Our work builds on these foundations and takes a step toward connecting semantic models with reinforcement learning theory. Rather than assuming that environment-provided signals are always available or reliable, we study when inferred semantic rewards can act as effective surrogates. We identify structural properties, in particular monotonicity of trajectory returns and quasi-Markov structure, as central to their reliability. In contrast to RLHF, preference learning, or prior VLM-reward methods, our goal is not to construct new reward models but to characterize when inferred rewards are suitable for trajectory-based policy gradient methods such as PPO and REINFORCE, and when they are fundamentally incompatible with value-based approaches such as

DQN. We analyze which algorithmic families are most compatible with such signals, and propose practical metrics to evaluate and guide reward inference in benchmark settings. In this way, our study complements prior empirical demonstrations by providing a conceptual framework that clarifies when semantic rewards succeed, when they fail, and how they might be improved.

## 3 INFERENCE-BASED RL FRAMEWORK (INFERL)

In standard reinforcement learning, an agent operates within a Markov Decision Process (MDP), defined by the tuple $(\mathcal{S}, \mathcal{A}, P, r, \gamma)$, where $r$ is a scalar reward provided by the environment. The agent aims to learn a policy $\pi(a|s)$ that maximizes the expected cumulative reward: $\mathbb{E}_\pi \left[ \sum_{t=0}^{\infty} \gamma^t r(s_t, a_t) \right]$.

This framework assumes that the environment provides a reliable and well-specified reward at each step. In practice, however, reward signals are often incomplete, noisy, or misaligned with the intended task, which can lead to undesirable behaviors even if the agent succeeds at maximizing the formal objective. We introduce a framework in which the reward function is inferred by the agent itself (Figure 2). We refer to this formulation as *Inference-Based Reinforcement Learning (InfeRL)*, highlighting its central departure from standard RL: environment-provided rewards are replaced with internally inferred signals. Throughout the paper, we use the term InfeRL to denote both the framework and the accompanying analysis.

**Reward Inference from Goals.** The inference-based formulation augments the MDP by introducing a goal space $\mathcal{G}$ and a reward inference function $f_{\text{inf}}$:

$$\mathcal{M}' = (\mathcal{S}, \mathcal{A}, P, \mathcal{G}, f_{\text{inf}}, \gamma).$$

Here, $\mathcal{G}$ denotes high-level task goals (e.g., language prompts or images), and $f_{\text{inf}} : (\tau, g) \to \mathbb{R}$ maps a trajectory prefix $\tau$ and a goal $g$ to a scalar reward. In general, $f_{\text{inf}}$ may take many forms depending on how goals are represented and how behavior is evaluated, ranging from classifiers to embedding-based similarity functions. This general structure captures the idea that the agent's reward is not provided by the environment but instead inferred through alignment with the specified goal.

It is important to note that in this formulation, the reward inference function $f_{\text{inf}}$ is part of the *agent*, not the environment. While in practice we instantiate $f_{\text{inf}}$ with a pretrained perceptual model (e.g., CLIP), conceptually this mechanism belongs to the agent's design and policy learning process. The environment does not provide inferred rewards; rather, the agent generates them internally by evaluating its own behavior against the specified goal. This distinction is crucial, as it separates the alignment challenge of reward inference from the external specification of the environment. This distinction is important in our later analysis of monotonicity and quasi-Markov structure, where the suitability of the inferred reward depends on the agent design rather than the environment.

To make the discussion concrete, we adopt a commonly used instantiation that has appeared in prior works and serves as the basis for our experiments. At the beginning of each episode, the agent is assigned a fixed goal prompt $g \in \mathcal{G}$. During interaction, it collects short trajectory segments $\tau_t = [I_{t-k+1}, \ldots, I_t]$ of observations, where $k$ denotes the segment length. In the simplest case $k = 1$, the inferred reward is based only on the most recent observation, while larger values of $k$ allow the reward to depend on a short history of frames. At every step, a pretrained perceptual model (e.g., CLIP (Radford et al., 2021)) computes alignment as

$$r_t = f_{\text{inf}}(\tau_t, g) = \alpha \cdot \cos\left(f_{\text{vision}}(\tau_t), f_{\text{text}}(g)\right),$$

where $\alpha$ is a scaling parameter. The functions $f_{\text{vision}}$ and $f_{\text{text}}$ embed short trajectories and goal descriptions into a shared representation space, and their cosine similarity serves as the inferred reward. This instantiation grounds the abstract framework in a concrete implementation while still allowing us to analyze the broader theoretical properties of inference-based rewards. At the same time, because the inferred reward depends on trajectory fragments rather than solely on the current state and action, it does not strictly satisfy the Markov property, making it a quasi-Markov signal whose effect on policy gradient and value-based methods differs in predictable ways.

**Goal-Conditioned Policies.** We condition the policy on the embedding of the goal, $u = f_{\text{text}}(g)$, which allows a single agent to generalize across tasks and instructions: $\pi(a_t \mid s_t, u)$. This enables the agent to adapt to new instructions at test time without retraining, as new goals can be incorporated directly through the reward inference mechanism. Importantly, the goal affects learning only through the inferred reward and its embedding, not through environment-provided supervision.

**Interpretation.** Inference-Based RL (InfeRL) preserves the trial-and-error structure of reinforcement learning but replaces externally specified rewards with semantically inferred signals. By grounding reward in a goal specification, this framework provides a formal lens on recent attempts to use pretrained models as reward providers. It also highlights the central question: under what conditions do such inferred rewards behave like reliable substitutes for environment-defined objectives? The rest of this paper develops theoretical criteria, algorithmic implications, and empirical evaluations to answer this question. In later sections we show that order-preserving monotonicity governs the suitability of these signals for trajectory-based policy gradient methods such as REINFORCE and PPO, while quasi-Markov structure plays a key role for value-based methods.

**Monotonicity and the Markov Property.** A central requirement for inference-based rewards to serve as effective surrogates is that they preserve the ordering of trajectories with respect to task performance. Formally, we use the order-preserving formulation: if $\hat{R}(\tau_1) \leq \hat{R}(\tau_2)$ then $R(\tau_1) \leq R(\tau_2)$. This monotonicity condition ensures that optimizing with respect to the inferred signal leads to policies that also improve true task outcomes. A related but distinct issue is whether the inferred reward satisfies the Markov property, depending only on the current state and action rather than the full trajectory. While this property often fails for inference-based signals, the resulting quasi-Markov structure helps explain why some algorithmic families are more robust than others. Our subsequent analysis examines when monotonicity holds, how to measure it in practice, and what algorithmic strategies can mitigate the absence of these assumptions.

## 4 THEORETICAL ANALYSIS OF INFERENCE-BASED REWARDS

Inference-based reinforcement learning departs from the standard MDP formulation by replacing externally defined rewards with signals inferred from a perceptual model. This modification raises fundamental theoretical questions: under what conditions can such inferred rewards still support effective learning, and what assumptions are required for different classes of algorithms? We identify two key properties, monotonicity and the Markov assumption, that play distinct roles in determining the reliability of learning under inferred rewards.

### 4.1 MONOTONICITY OF INFERRED REWARDS

**Definition.** A central requirement for inference-based rewards to serve as reliable surrogates is that they preserve the ordering of trajectories with respect to true task performance. Let $R(\tau)$ denote the true return of a trajectory $\tau$, and $\hat{R}(\tau)$ the inferred return produced by $f_{\text{inf}}$. We use an order-preserving formulation of monotonicity: for any two trajectories $\tau_1, \tau_2$,

$$\hat{R}(\tau_1) \leq \hat{R}(\tau_2) \quad \Longrightarrow \quad R(\tau_1) \leq R(\tau_2), \tag{1}$$

Equivalently, whenever the inferred return ranks $\tau_2$ at least as high as $\tau_1$, the true return does not contradict this ordering. This condition ensures that the inference function $f_{\text{inf}}$ preserves the relative ranking of behaviors, even if the inferred rewards differ in scale or contain noise. When monotonicity holds, optimizing with respect to $\hat{R}(\tau)$ will guide the agent toward trajectories that also improve the true return $R(\tau)$. By contrast, violations of monotonicity can result in reward misalignment, where $f_{\text{inf}}$ favors trajectories that appear successful under the inferred signal but in fact degrade true task performance.

At the framework level, monotonicity formalizes when internally inferred signals can safely replace environment-provided rewards. It highlights that the key property is not the absolute accuracy of the reward values, but their ability to order trajectories consistently with the underlying task objective. This makes monotonicity a central criterion for analyzing the effectiveness of inference-based reinforcement learning.

**Measuring Monotonicity.** Monotonicity can be quantified by comparing the ordering of trajectories under the true return $R(\tau)$ and the inferred return $\hat{R}(\tau)$. Concretely, given a collection of trajectories $\{\tau_i\}_{i=1}^N$, one may compute: (i) *pairwise agreement*, the fraction of trajectory pairs $(\tau_i, \tau_j)$ for which the ordering induced by $R(\tau)$ and $\hat{R}(\tau)$ is consistent; (ii) *rank correlation* measures such as Kendall's $\tau$ or Spearman's $\rho$. Together, these metrics provide a practical monotonicity score and guide design choices such as prompt specification or trajectory representation.

## 4.2 THEORETICAL ROLE OF MONOTONICITY IN POLICY IMPROVEMENT

Monotonicity ensures that optimizing the inferred return remains aligned with the true task objective. When rewards are inferred from a vision–language model, their scale and noise properties may differ significantly from the environment-provided reward. What matters for learning, however, is whether the inferred return preserves the ordering of trajectories under the true return. If an update increases the expected inferred return, it should also increase the expected true return. This requirement does not rely on Markovian assumptions and depends only on the relative ranking of complete trajectories.

Formally, under the order-preserving condition in equation 1, the inferred reward never contradicts the ordering of trajectories induced by the true reward.

**Lemma.** *Let $\hat{R}(\tau)$ be an inferred return satisfying the order-preservation condition in (1). Any policy update that increases the expected inferred return $\mathbb{E}_{\tau \sim \pi}[\hat{R}(\tau)]$ also increases the expected true return $\mathbb{E}_{\tau \sim \pi}[R(\tau)]$.*

**Proof sketch.** Policy gradient methods increase the probability of trajectories with larger $\hat{R}(\tau)$. By monotonicity, such trajectories must also have larger $R(\tau)$. Thus updates that improve $\mathbb{E}[\hat{R}(\tau)]$ necessarily improve $\mathbb{E}[R(\tau)]$. This argument operates at the level of full trajectories and does not require $\hat{r}_t$ to be Markovian.

This lemma applies directly to REINFORCE, which performs unbiased Monte Carlo policy gradient updates using complete trajectory returns. While PPO includes a value baseline, the critic is used only for variance reduction. As a result, PPO inherits the robustness properties of trajectory-level methods and remains effective even when inferred rewards are non-Markovian. In contrast, value-based methods rely on Markovian per-step rewards and may therefore propagate inconsistent updates when $\hat{r}_t$ depends on short temporal windows.

## 4.3 MARKOV PROPERTY OF INFERRED REWARDS

**Definition.** In the standard MDP formulation, the reward function is assumed to satisfy the *Markov property*, depending only on the current state and action. Formally, $r(s_t, a_t)$ is determined solely by $(s_t, a_t)$, independent of the past trajectory. This assumption underlies many theoretical guarantees in reinforcement learning, particularly for algorithms based on dynamic programming and Bellman equations. In inference-based reinforcement learning, however, rewards are produced by an inference function $f_{\text{inf}}$ that often depends on trajectory fragments $\tau_t = [I_{t-k+1}, \ldots, I_t]$ rather than a single state-action pair. This design enables richer semantic evaluation, but it means that the inferred reward generally does not satisfy the strict Markov property. As a result, standard assumptions used by Q-learning style methods may break down, since the reward cannot be decomposed step by step without introducing bias. To understand how Markovian structure may be approximated in practice, we introduce the notion of temporal consistency.

**Temporal Consistency.** There exists $k \geq 1$ such that for all $(s_t, a_t)$, $\hat{r}(s_t, a_t) = f(r(s_t, a_t), s_{t:t+k})$, where $s_{t:t+k}$ denotes a trajectory segment of length $k$, and $f$ is a deterministic mapping. This assumption states that the inferred reward $\hat{r}$ becomes Markovian, or quasi-Markovian, when conditioned on a bounded temporal window. A single state-action pair may be insufficient to determine reward meaningfully, but a short segment can disambiguate states that appear similar locally yet differ semantically. Without this property, the inferred reward may collapse distinct outcomes, creating systematic misalignment between $\hat{r}$ and the true reward $r$.

**Quasi-Markov Rewards via Augmented MDPs.** If the inferred reward depends on a finite trajectory window, one can define an *augmented MDP* where the same reward is Markovian, allowing standard RL guarantees to apply.

**Past-window form.** An inferred reward is $k$-quasi-Markov if there exists a function $g : \mathcal{S}^k \times \mathcal{A} \times \mathcal{U} \to \mathbb{R}$ such that $\hat{r}_t = g\big((s_{t-k+1}, \ldots, s_t), a_t, u\big)$, where $u \in \mathcal{U}$ is a fixed goal or condition (e.g., a language embedding).

**Future-window form.** An inferred reward is $k$-quasi-Markov if there exists a function $h : \mathcal{S}^{k+1} \times \mathcal{A} \times \mathcal{U} \to \mathbb{R}$ such that $\hat{r}_t = h\big((s_t, \ldots, s_{t+k}), a_t, u\big)$.

**Practical Implications.** The past-window form computes rewards from a short history (e.g., a clip ending at $t$). The future-window form uses a short look-ahead (e.g., verifying stability over the next $k$ frames), which can be made causal by delaying the reward by $k$ steps. Quasi-Markov structure provides a practical way to reconcile inference-based rewards with RL theory. In our experiments, varying the temporal window size $k$ directly modulates this quasi-Markov structure and has a pronounced effect on value-based methods such as DQN, while trajectory-based policy gradient methods remain comparatively stable.

### 4.4 Algorithmic Implications

A central question in reinforcement learning with inferred rewards is whether existing algorithms can still guarantee policy improvement when the reward function is imperfect. We focus on two widely used families: trajectory-based policy gradient methods and step-wise value-based methods.

**Policy Gradient Methods.** Policy gradient algorithms optimize the expected return

$$J(\pi) = \mathbb{E}_{\tau \sim \pi}\left[R(\tau)\right] = \mathbb{E}_{\tau \sim \pi}\left[\sum_{t=0}^{T} r_t\right],$$

where $\tau = (s_0, a_0, \ldots, s_T, a_T)$ denotes a trajectory. The policy gradient theorem gives

$$\nabla_\theta J(\pi_\theta) = \mathbb{E}_{\tau \sim \pi_\theta}\left[\sum_{t=0}^{T} \nabla_\theta \log \pi_\theta(a_t|s_t)\, \hat{R}(\tau)\right],$$

where $\hat{R}(\tau)$ may be an inferred return. If the inferred return $\hat{R}(\tau)$ preserves the ordering of true returns $R(\tau)$ (i.e., monotonicity holds), then the gradient direction remains positively correlated with improvements in $R(\tau)$. Thus, even if $\hat{R}(\tau)$ is noisy or non-Markovian, trajectory-level monotonicity suffices for consistent improvement under policy gradient updates.

**Q-Learning and Bellman Updates.** Q-learning algorithms, by contrast, rely on the Bellman equation

$$Q^\pi(s, a) = \mathbb{E}\left[r(s, a) + \gamma \max_{a'} Q^\pi(s', a')\,\Big|\, s, a\right],$$

which assumes that the reward $r(s, a)$ is a Markovian function of the current state-action pair. If rewards are inferred from trajectory fragments, i.e., $\hat{r}_t = f_{\text{inf}}(\tau_t, g)$, with $\tau_t = (s_{t-k+1}, \ldots, s_t)$ or $(s_t, \ldots, s_{t+k})$, then $\hat{r}_t$ may depend on history or future context. In such cases, two identical state-action pairs $(s_t, a_t)$ may yield different inferred rewards depending on surrounding context, violating the Markov property. This breaks the Bellman recursion and leads to biased targets even when trajectory-level monotonicity holds. Consequently, value propagation may mis-rank state-action pairs and destabilize learning. Augmenting the state with a temporal window (i.e., increasing $k$) can partially restore quasi-Markov structure and improve DQN performance, but our results indicate that value-based methods remain more sensitive to violations of Markovian assumptions than trajectory-based policy gradient methods.

**Implications.** This analysis highlights an asymmetry. Policy gradient methods require only trajectory-level monotonicity, making them robust to non-Markovian or noisy inferred rewards. Q-learning methods, however, require both monotonicity and strict (or quasi-) Markovianity of per-step rewards to ensure stability. This explains why trajectory-based methods tend to perform better in practice under inferred rewards, while Q-learning style algorithms are more fragile.

## 5 Experiments

### 5.1 Experimental Setup

We evaluate InfeRL across classic control and MuJoCo environments, including variants of CartPole with symbolic visual cues, as well as Ant and Walker2D tasks that require more complex motor coordination. At each timestep, a pretrained VLM (e.g., CLIP) provides inferred rewards by aligning short trajectory windows with natural language goals, replacing environment-supplied signals. We compare three algorithmic families: trajectory-level policy gradient methods (REINFORCE and

PPO) and a step-wise value-based method (DQN), to probe the algorithmic asymmetries identified in our theory. We also vary the temporal window size $k$ used for reward inference to study how quasi-Markov structure influences learning dynamics. Full details of environments, prompts, reward inference, and evaluation protocols are provided in Appendix B.

## 5.2 MEASURING MONOTONICITY OF INFERRED REWARDS

To quantify the reliability of inferred rewards, we compute pairwise agreement, Kendall's $\tau$, and Spearman's $\rho$ (Virtanen et al., 2020) between rankings induced by inferred returns $\hat{R}(\tau)$ and true returns $R(\tau)$. Ground-truth returns are estimated from trajectories generated by a random policy, ensuring a diverse set of behaviors for evaluation. However, for locomotion tasks, random rollouts alone do not adequately span both successful and failing behaviors, since many desirable outcomes such as stable walking rarely occur under random actions. In contrast, using fully trained policies would bias the dataset toward high-return trajectories and would severely underrepresent failure cases. To obtain a balanced and representative set of trajectories, we supplement random rollouts with trajectories sampled from partially trained PPO policies. For Walker2D and Ant, we use policies at 500K timesteps, which corresponds to one quarter of the standard two million timesteps required for convergence. This choice provides access to trajectories that include early failures, partially successful attempts, and emerging locomotion skills. The resulting mixed dataset captures a wide spectrum of behaviors and provides a more reliable basis for estimating monotonicity.

Table 1 reports monotonicity scores across four environments, each tested with three different natural language prompts. These prompts were chosen to represent both aligned goals (e.g., "upright walking") and misaligned ones (e.g., "collapsed robot"), allowing us to assess whether monotonicity can distinguish suitable reward specifications from poor ones.

Table 1: Monotonicity results across environments and prompts. Values report pairwise agreement, Kendall's $\tau$, and Spearman's $\rho$ between rankings induced by true and inferred returns.

| Environment | Prompt | Pairwise Agr. | Kendall's $\tau$ | Spearman's $\rho$ |
|---|---|---|---|---|
| CartPole | keep the pole upright. | 1.000 | 0.981 | 0.999 |
| | The pole is nearly vertical (upright), and the cart is near the center. | 1.000 | 0.983 | 0.999 |
| | The pole has fallen over, lying flat instead of upright. | 1.000 | 0.979 | 0.998 |
| InvPend | keep the pole upright. | 1.000 | 0.939 | 0.990 |
| | The pole is nearly vertical (upright), and the cart is near the center. | 1.000 | 0.930 | 0.987 |
| | keep the pole downward. | 1.000 | 0.925 | 0.984 |
| *Policy →* | | Random / Partial | Random / Partial | Random / Partial |
| Walker2d | A robot walking upright steadily. | 0.668 / 0.929 | 0.337 / 0.859 | 0.474 / 0.966 |
| | A two-legged robot walking steadily to the right in an upright posture. | 0.669 / 0.943 | 0.339 / 0.886 | 0.473 / 0.979 |
| | A two-legged robot collapsed on the ground, lying sideways and not walking. | 0.675 / 0.938 | 0.351 / 0.876 | 0.489 / 0.974 |
| Ant | A four-legged robot walking and balanced. | 0.306 / 0.794 | -0.388 / 0.587 | -0.556 / 0.773 |
| | An ant robot walking steadily forward, with coordinated motion. | 0.336 / 0.769 | -0.328 / 0.538 | -0.469 / 0.700 |
| | A four-legged robot collapsed on the ground, lying flat and not walking. | 0.271 / 0.797 | -0.459 / 0.595 | -0.657 / 0.766 |

**Analysis.** In the **CartPole** and **InvertedPendulum** domains, all prompts exhibit near perfect monotonicity, with Kendall's $\tau$ greater than $0.92$ and Spearman's $\rho$ greater than $0.98$. This reflects the simplicity of these tasks, where semantic descriptions align cleanly with environment dynamics. Even prompts referring to the pole being fallen or downward produce consistent rankings, showing that monotonicity is easy to satisfy in well structured environments.

In **Walker2D**, results based on random trajectories are weaker but still meaningful, with $\tau$ around $0.33$ to $0.35$. Despite the complexity of bipedal locomotion, the inferred rewards distinguish between successful walking and collapsed robots, indicating that monotonicity analysis can highlight misaligned prompts even when absolute correlations are modest. When evaluated on trajectories from a partially trained PPO policy, monotonicity increases substantially. Here partial refers to full trajectories generated by a policy that has learned some locomotion. These rollouts contain both failures and emerging gait patterns, which provide clearer semantic differences aligned with the prompts, leading to higher agreement with true returns.

In **Ant**, the most challenging domain, all prompts yield negative correlations under random rollouts. Nevertheless, the relative ordering remains informative since collapsed behaviors are consistently ranked worse than walking related ones. Under partial trajectories, monotonicity becomes positive because the dataset contains both collapse and meaningful motion, allowing the inferred reward to distinguish behaviors more reliably.

Overall, these results suggest that monotonicity analysis provides a useful lens for assessing whether inferred rewards are suitable for learning in specific settings. It helps distinguish between prompts that yield reliable signals and those likely to mislead the agent. Comparing random and partial datasets shows consistent trends, with partial trajectories offering clearer structure and higher agreement, supporting the robustness of monotonicity as a diagnostic. Additional analysis examining the effect of language prompt phrasing is provided in Table 3, showing that prompt-level monotonicity strongly predicts PPO performance.

### 5.3 POLICY LEARNING UNDER INFERRED REWARDS

**Comparison of Algorithms** We compare three algorithms under inference-based rewards: REINFORCE as a pure trajectory-level policy gradient method, PPO as a variance-reduced policy gradient method, and DQN as a step-wise value-based method. The results across three variants of the CartPole environment are summarized in Table 2. In the CartPole-Base setting, both REINFORCE and PPO achieve reliable learning, consistent with the observation that this environment admits a well-defined ground-truth reward and that the inferred signal exhibits strong monotonicity with true returns. The success of both trajectory-based methods reflects the theoretical prediction that monotonicity alone is sufficient for policy gradient improvement,

Table 2: Episodes and timesteps required for REINFORCE, PPO, and DQN to solve tasks across CartPole variants. Trajectory-based methods converge reliably under inferred rewards, while DQN struggles when Markov structure is weak.

| Environment | REINFORCE | | PPO | | DQN | |
|---|---|---|---|---|---|---|
| | Ep. | Steps | Ep. | Steps | Ep. | Steps |
| CartPole-Base | **254** | 45K | 280 | **12K** | 2024 | 58K |
| CartPole-FireWater | 450 | 23K | **300** | **14K** | 2800 | 42K |
| CartPole-MultiCue | 780 | 41K | **360** | **25K** | 3100 | 60K |

even when the inferred reward is not strictly Markovian. By contrast, DQN also succeeds in this setting but requires significantly more interaction, suggesting that near-Markovian structure in the CartPole-Base reward makes it an unusually favorable case for value-based learning. In the CartPole-FireWater variant, both REINFORCE and PPO remain effective, with REINFORCE solving the task in 450 episodes and PPO in about 300 episodes (14K steps). DQN requires far more interaction. A similar pattern appears in CartPole-MultiCue, where REINFORCE converges in 780 episodes and PPO in 360 episodes, while DQN again requires substantially more training. These results reinforce the theoretical asymmetry: trajectory-based methods depend only on monotonicity of complete trajectories, whereas DQN additionally requires per-step Markovianity, which is weakened by the window-based inferred reward.

Together, these experiments provide empirical validation of the theoretical claims. Trajectory-based policy gradient methods can reliably exploit monotonic but non-Markovian rewards, whereas step-wise Bellman-based methods become unstable, requiring substantially more samples or failing to converge. The consistency between the monotonicity analysis and observed learning behavior supports the view that monotonicity serves as a useful diagnostic for algorithmic choice under inference-based reinforcement learning. The similar performance patterns of REINFORCE and PPO further highlight that the key requirement for success under inferred rewards is trajectory-level order preservation rather than any specific value-learning structure.

Overall, our experiments indicate that monotonicity offers a useful diagnostic for evaluating reward reliability, that trajectory-based methods such as REINFORCE and PPO align better with inferred rewards than step-wise methods like DQN, and that instruction-driven behaviors showcase both the promise of semantic rewards and the limitations arising from ambiguity and non-Markov dependencies. Further details are provided in Appendix C.5.

## 5.4 EFFECT OF FRAME STACKING ON MARKOVITY OF INFERRED REWARDS

To evaluate whether increasing temporal context improves the Markovity of the inferred reward, we vary the number of stacked frames used as input to the VLM-based reward function. Frame stacking provides a short history at each timestep and therefore reduces ambiguity in the inferred reward, which makes the per-step signal closer to a Markov function of the current observation window. Figure 1 summarizes the number of timesteps required by REINFORCE, PPO, and DQN to solve the CartPole-Base task for stack sizes $k \in \{1, 2, 4\}$. The trajectory-based methods, REINFORCE and PPO, show only mild sensitivity to the stack size, which is consistent with our theory that

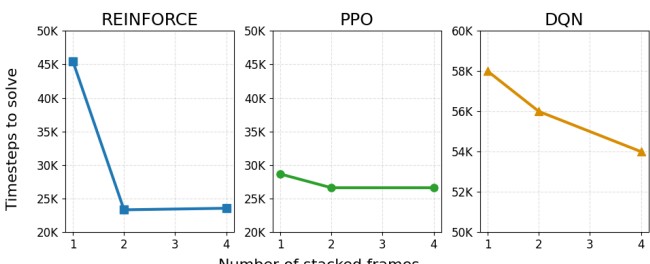

Figure 1: **Effect of frame stacking on Markovity of inferred rewards (CartPole-Base).** More stacked frames improve DQN performance by making the inferred reward more Markovian, while REINFORCE and PPO remain largely insensitive. This illustrates the asymmetry between trajectory-based and Bellman-based methods under non-Markovian rewards.

policy gradient methods rely on trajectory-level monotonicity rather than strict per-step Markovity. In contrast, DQN exhibits a clear benefit from increased temporal context. As $k$ increases, DQN requires fewer timesteps to solve the task, reflecting that a larger observation window restores partial Markov structure and reduces the inconsistencies introduced by window-based reward inference. Overall, these results support the asymmetry predicted by our theoretical analysis. Frame stacking has limited impact on trajectory-based algorithms but substantially improves the stability of value-based methods by making the inferred reward closer to a per-step Markov signal.

## 5.5 EFFECT OF LANGUAGE PROMPT

Table 3 shows that PPO solves the CartPole-Base task consistently across prompts that express the same goal. Solve times vary only slightly, which is reflected in their high monotonic-

Table 3: PPO performance for solving the CartPole-Base task under different natural language prompts.

| ID | Prompt (Kendall's $\tau$) | | Timesteps | Episodes |
|---|---|---|---|---|
| 1 | *keep the pole upright.* | ($\tau = 0.980$) | 29K | 317 |
| 2 | *The pole is nearly vertical (upright), and the cart is near the center of the track.* | ($\tau = 0.983$) | 27K | 294 |
| 3 | *The pole is nearly vertical (upright), and the cart is near the center.* | ($\tau = 0.981$) | 27K | 298 |

ity scores ($\tau = 0.980, 0.983, 0.981$), indicating that each prompt induces nearly identical trajectory rankings. These results suggest that in simple environments such as CartPole, inferred rewards are robust to small linguistic variations. In more complex settings, where visual dynamics are richer, prompt phrasing may have a larger effect on monotonicity and learning behavior.

## 6 CONCLUSION

We studied InfeRL, a framework that replaces environment-provided rewards with semantic signals inferred from vision–language models. Our analysis highlighted monotonicity and (quasi-)Markovianity as key properties shaping when such rewards support effective learning. Experiments across control domains showed that monotonicity correlates with learning outcomes, policy gradient methods such as REINFORCE and PPO are more robust than value-based methods such as DQN, and natural language prompts can guide both standard and novel behaviors. Additional experiments also indicate that prompt phrasing can influence reward monotonicity, though simple tasks such as CartPole remain largely robust. These results suggest that viewing reinforcement learning through the lens of reward inference provides a principled path toward aligning agents with high-level goals while underscoring the need for careful prompt design and robust inference mechanisms.

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

**Standard Reinforcement Learning (MDP)**

*Formalism:*

$$\mathcal{M} = (\mathcal{S}, \mathcal{A}, P, r, \gamma)$$

- $\mathcal{S}$: state space
- $\mathcal{A}$: action space
- $P(s'|s, a)$: transition dynamics
- $r(s, a)$: reward from environment
- $\gamma$: discount factor

*Objective:*

$$\max_{\pi} \; \mathbb{E}_{\pi} \left[ \sum_{t=0}^{\infty} \gamma^t r(s_t, a_t) \right]$$

*Reward:* Environment-provided
*Policy:* $\pi(a_t|s_t)$

**Inference-Based RL (InfeRL)**

*Formalism:*

$$\mathcal{M}' = (\mathcal{S}, \mathcal{A}, P, \mathcal{G}, f_{\text{inf}}, \gamma)$$

- $\mathcal{S}$: state space
- $\mathcal{A}$: action space
- $P(s'|s, a)$: transition dynamics
- $\mathcal{G}$: goal space (e.g., text)
- $f_{\text{inf}}(\tau, g)$: inferred reward
- $\gamma$: discount factor

*Objective:*

$$\max_{\pi} \; \mathbb{E}_{\pi} \left[ \sum_{t=0}^{\infty} \gamma^t f_{\text{inf}}(\tau_t, g) \right]$$

*Reward:* Inferred by agent
*Policy:* $\pi(a_t|s_t, f_{\text{text}}(g))$

Figure 2: Inference-Based RL (InfeRL) replaces externally defined rewards with internally inferred signals based on semantic alignment between behavior and goals.

## A  STANDARD MDP VS INFERENCE-BASED RL (INFERL)

Figure 2 summarizes the structural difference between the standard MDP formulation and the InfeRL framework used in this work. In a conventional MDP, the reward is a property of the environment and depends only on the current state and action. This design assumes that the task objective is known in advance and can be encoded directly into a Markovian reward function.

In InfeRL, the reward is produced internally by the agent through an inference mechanism that evaluates short trajectory segments against a goal specification. This shift has two important implications. First, the reward signal is no longer guaranteed to be Markovian since it may depend on a short history of observations. Second, the reward becomes a design component of the agent rather than a fixed part of the environment, which allows goal specifications to be changed without modifying the underlying dynamics.

These distinctions clarify why standard theoretical guarantees for value-based methods may not apply directly under inferred rewards, and why policy gradient methods, which operate on full trajectories, tend to be more robust. At the same time, the InfeRL formulation highlights the benefit of using rich semantic models to express goals in a flexible and natural way, enabling learning in settings where explicit reward engineering is difficult or infeasible.

## B  EXPERIMENT SETUP

To empirically validate our theoretical analysis, we design experiments that evaluate how inference-based rewards enable agents to acquire meaningful behaviors from natural language prompts and visual feedback, without relying on environment-supplied rewards. Our objectives are threefold: to measure whether inferred rewards preserve monotonicity with ground-truth returns, to compare the robustness of trajectory-based and step-wise algorithms under inferred rewards, and to test whether semantic prompts support novel instructed behaviors that are difficult to specify with handcrafted reward functions.

**Environments** We conduct experiments in three control domains of increasing complexity: CartPole, MuJoCo Ant, and MuJoCo Walker2D. The CartPole domain serves as a controlled setting where we introduce progressively richer variants (see Figure 3). In CartPole-Base, the agent balances a pole on a moving cart. CartPole-FireWater augments the background with symbolic cues, a fire icon on the left and a water droplet on the right, enabling instructions such as "move toward water" or "avoid

fire." CartPole-MultiCue extends this idea further, adding multiple cues such as umbrellas and clouds to support more abstract and context-dependent prompts like "stay under the umbrella while avoiding hazards."

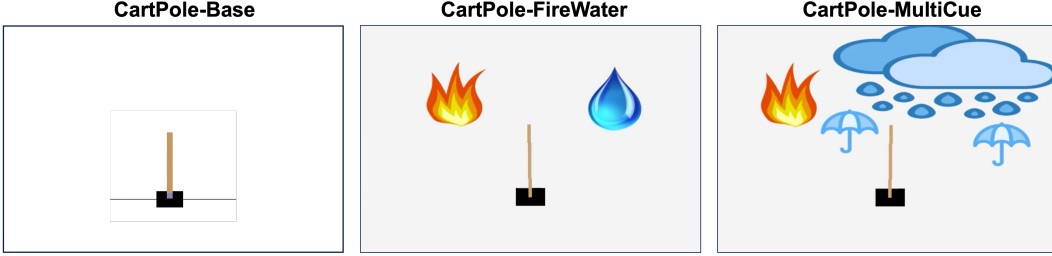

Figure 3: **Variants of the CartPole environment used in our experiments.** From left to right: (1) **CartPole-Base**: the standard task where the agent balances a pole on a cart. (2) **CartPole-FireWater**: background includes a fire icon on the left and a water droplet on the right, enabling directional prompts. (3) **CartPole-MultiCue**: background includes fire, water droplets, umbrellas, and clouds, supporting more complex instructions such as "stay under the umbrella" or "avoid fire." These variations preserve the original task dynamics while introducing symbolic visual cues that enable natural language instruction grounding. They are designed to evaluate the agent's ability to infer rewards from semantics, generalize across goal specifications, and follow increasingly abstract or context-dependent instructions.

The Ant environment offers a substantially more complex control challenge with high-dimensional action and observation spaces. We consider two tasks: Ant-Balance, in which the agent must rotate in place while maintaining stability, and Ant-Rotate, where the goal is to spin rapidly in place without forward locomotion. The latter illustrates how simple natural language prompts can express behaviors that are difficult to encode via handcrafted rewards. Finally, Walker2D provides another high-dimensional setting where the agent is instructed to "walk while remaining upright."

These domains preserve the underlying dynamics of their respective environments while enabling more expressive, interpretable goals. This makes them ideal testbeds for examining how well inference-based rewards capture task intent.

**Goal Prompts** Each environment is paired with natural language goal specifications that describe the desired behavior (Table 4). Prompts are designed to be semantically meaningful yet sufficiently underspecified to highlight potential ambiguities. For example, CartPole-Base is defined by "the pole remains upright and the cart stays near the center," while Ant-Rotate uses "a four-legged ant robot spins in place while staying balanced." We also test robustness to prompt variations by rephrasing instructions or altering emphasis.

Table 4: Natural language goal specifications used for reward inference across different environment settings.

| Environment | Setting | Task Type | Goal Specification |
|---|---|---|---|
| CartPole | Base | Single-objective | The pole is nearly vertical (upright), and the cart is near the center of the track. |
| CartPole | FireWater | Multi-objective | A cart with an upright pole is positioned directly under a red and yellow fire icon, far away from the blue water droplet. |
| CartPole | MultiCue | Multi-objective + Complex | A cart with an upright pole is positioned directly under a red and yellow fire icon, far away from the blue water droplet. |
| | | Multi-objective + Complex | The pole is upright and stable, with both the cart and pole positioned under the right umbrella, far from the fire and out of the rain. |
| MuJoCo Ant | Balance | Single-objective | A four-legged robot walking and balanced. |
| MuJoCo Ant | Rotate | Novel behavior | A four-legged ant robot spinning rapidly in place, staying centered and balanced. |
| MuJoCo Walker2D | Walk | Single-objective + Ambiguous | A robot walking upright steadily. |

**Reward Inference Mechanism** At each timestep, a pretrained vision–language model such as CLIP (Radford et al., 2021) encodes a short trajectory window into an embedding. This embedding

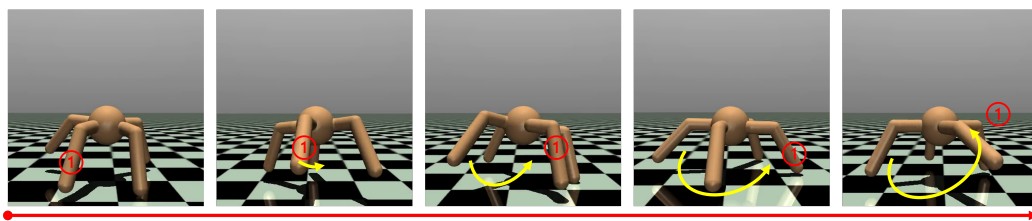

**Ant Rotate Goal Specification:**
A four-legged ant robot spinning rapidly in place, staying centered and balanced.

Figure 4: **Ant-Rotate behavior guided by a language-specified goal.** A sequence of frames showing the Ant agent rotating counterclockwise in place. Red circles mark a front leg for orientation; yellow arrows indicate the direction of rotation. The behavior is learned solely from natural language-based reward inference, without handcrafted shaping or environment-provided rewards (video in supplementary).

is compared with that of the goal prompt, and their cosine similarity defines the inferred reward. The environment's native reward function is ignored during training. Importantly, the inference mechanism is part of the agent's design rather than the environment specification, reinforcing the conceptual distinction at the heart of InfeRL: rewards are inferred internally rather than externally supplied.

**Reinforcement Learning Algorithms** We compare two widely used algorithmic families that embody the asymmetry highlighted in our theoretical analysis. Proximal Policy Optimization (PPO) (Schulman et al., 2017) serves as a trajectory-based policy gradient method, which can tolerate non-Markov rewards as long as monotonicity is preserved at the trajectory level. Deep Q-Networks (DQN) (Mnih et al., 2015) represent step-wise value-based methods, which require both monotonicity and Markovian structure in per-step rewards. Implementations are based on Stable-Baselines3 (Raffin et al., 2021) and CleanRL (Huang et al., 2022), with no algorithmic modifications. The cosine similarity produced by the vision–language model is passed directly as the per-step reward, demonstrating that InfeRL integrates seamlessly with standard RL pipelines.

**Evaluation Metrics** We evaluate agents along two complementary axes. First, we measure monotonicity by comparing the ordering of trajectories under inferred and ground-truth rewards, reporting pairwise agreement and rank correlation (Kendall's $\tau$, Spearman's $\rho$). Second, we assess learning performance both quantitatively, via episode returns under PPO and DQN, and qualitatively, through manual inspection of trained policies across multiple rollouts to determine whether behaviors align with the intended natural language goals. Each experiment is repeated across five random seeds, with ten rollouts per trained policy. This dual evaluation allows us to probe both alignment fidelity and algorithmic robustness.

## C   ADDITIONAL RESULTS

### C.1   INSTRUCTION-DRIVEN BEHAVIORS

We first evaluate whether InfeRL can induce a novel behavior in the **MuJoCo Ant** environment. The agent receives the goal description: *"a four-legged ant robot spinning in place while staying balanced."* This objective requires a significant departure from the default locomotion typically observed in Ant tasks, instead demanding symmetric leg movements that achieve rotation without translation.

As shown in Figure 4, the agent successfully learns to rotate in place while maintaining balance. Frame sequences illustrate consistent angular displacement, with leg markers and arrows confirming stable counterclockwise spinning. Importantly, this outcome is achieved without handcrafted reward shaping or explicit motion specification, relying solely on the inferred reward signal derived from a pretrained vision–language model.

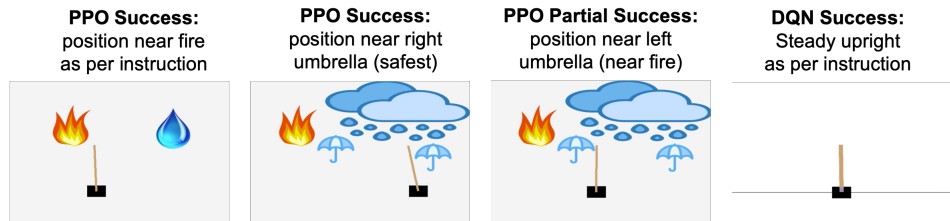

Figure 6: **CartPole instruction following with PPO and DQN.** Visualizations of final agent behaviors under different prompt types and environments. **Left to Right:** (1) PPO-trained agent in the FireWater setting learns to position itself near the fire icon, consistent with the provided instruction. (2) In the MultiCue environment, the PPO agent successfully navigates to the rightmost umbrella, avoiding fire and rain, as specified by the goal. (3) A partial success in the MultiCue environment, where the agent stops near the left umbrella, satisfying some but not all constraints. (4) DQN agent, operating with a **discrete action space**, is evaluated on the base CartPole setup and learns to stay upright and centered in accordance with the instruction. These results illustrate InfeRL's ability to support multi-objective goals under both continuous (PPO) and discrete (DQN) control regimes (video in supplementary).

This result demonstrates that inference-based rewards can support the acquisition of non-default, instruction-driven behaviors that are difficult to express through standard environment rewards. It highlights the flexibility of InfeRL in aligning agent behavior with semantically specified goals.

We further test generalization in the **MuJoCo Walker2D** environment, where the agent is instructed with the prompt: *"a robot walking upright steadily."* As illustrated in Figure 7, the agent learns a stable gait that preserves balance and posture but consistently moves backward.

This outcome underscores both the promise and the limitations of natural language reward inference. On one hand, the instruction successfully drives upright walking without task-specific reward engineering. On the other hand, the absence of explicit directional cues allows the agent to adopt a behavior that is semantically consistent with the language model's interpretation but misaligned with human expectations. Such cases emphasize the importance of precise goal specification and connect directly to our theoretical analysis: vague or underspecified prompts risk violating monotonicity and producing behaviors that, while interpretable, diverge from intended outcomes.

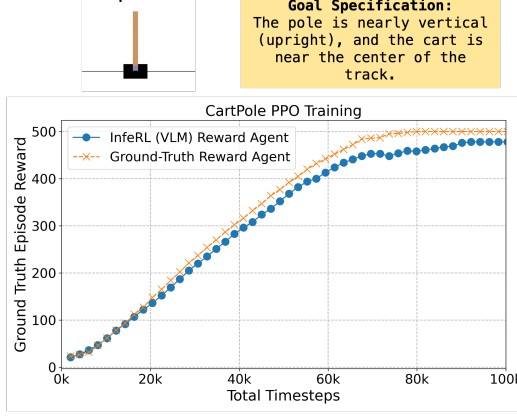

Figure 5: **CartPole PPO training results.** InfeRL PPO achieves performance comparable to a ground-truth reward agent, demonstrating that inferred rewards based on natural language goals can effectively guide policy learning. Performance is measured by episode length, corresponding to the agent's ability to maintain balance before termination.

## C.2 GENERALIZATION TO MULTI-OBJECTIVE AND COMPOSITIONAL GOALS

Beyond reproducing ground-truth rewards in standard control settings, we evaluate whether InfeRL can generalize to tasks requiring multi-objective and compositional reasoning. To this end, we consider two modified versions of the CartPole environment: **CartPole-FireWater** and **CartPole-MultiCue**. Both variants introduce symbolic visual cues that must be interpreted in conjunction with the pole-balancing objective, yielding natural language prompts that combine multiple constraints (see Table 4).

In CartPole-FireWater, the agent is instructed to position the cart near the fire icon and away from the water droplet, while also keeping the pole upright. In practice, PPO agents often succeed in moving toward the fire region, suggesting that the vision–language model correctly associates the fire symbol with the prompt. However, the need to maintain balance can conflict with positional goals, occasionally leading to instability or divergence when multiple constraints must be satisfied simultaneously.

In CartPole-MultiCue, the instruction specifies a richer objective involving fire, umbrellas, and rain. Here, we observe more consistent alignment with the intended goals: PPO agents frequently navigate toward the rightmost umbrella while avoiding fire and rain. Interestingly, partial successes also emerge, with the agent stopping under the left umbrella. This indicates that semantic similarity captures some but not all aspects of spatial relations among objects. One explanation is that umbrella and rain cues in this variant are visually distinct and semantically well-grounded in pretrained vision–language models, whereas the stylized water droplet in FireWater is less prototypical.

Figure 6 illustrates these qualitative outcomes. In both FireWater and MultiCue, PPO agents demonstrate the ability to follow composite instructions, though with varying levels of precision depending on the distinctiveness of visual cues.

Finally, to verify the framework's applicability beyond continuous control, we train a DQN Mnih et al. (2015) agent on the standard CartPole setting. The DQN agent successfully learns to keep the pole upright and the cart near the center, consistent with the prompt. This result indicates that InfeRL can operate in both continuous and discrete action spaces, while still supporting multi-objective goals when visual and linguistic cues are sufficiently clear.

## C.3 COMPARISON WITH GROUND-TRUTH REWARDS

To further validate the effectiveness of InfeRL, we compare PPO trained on inferred rewards with PPO trained on environment-provided ground-truth rewards. For the CartPole-Base environment, results show that PPO with inferred rewards achieves nearly identical performance to training with ground-truth rewards, confirming that semantic alignment from the vision–language model is sufficient to replicate the standard reward. Similar findings hold for the InvertedPendulum-v4 (the MuJoCo equivalent of CartPole), where episode length serves as the ground-truth reward signal. Learning curves for CartPole are presented in Figure 5, showing the close correspondence between ground-truth and inferred-reward training.

## C.4 WALKER2D WITH INFERRED REWARDS

A more interesting pattern emerges in Walker2D. Here, PPO with CLIP-based inferred rewards demonstrates faster early-stage learning compared to training with ground-truth rewards. Since the environment's ground-truth reward is unavailable in our setup, we report episode length as a proxy metric, which still provides a reasonable indication of task success. PPO with inferred rewards achieves an average episode length of 715 within 500K timesteps, compared to 476 under ground-truth rewards.

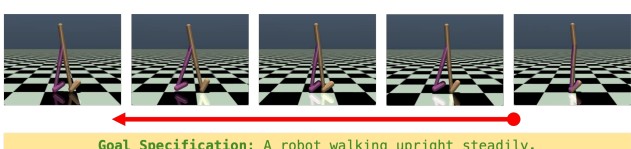

Figure 7: **MuJoCo Walker2D behavior under inferred reward.** A sequence of frames showing the agent walking upright steadily in the backward direction. The agent learns to maintain balance and upright posture but chooses to walk in reverse, highlighting a partial success and the impact of ambiguous language instructions.

One possible explanation for this difference is the prior knowledge embedded in CLIP, which may accelerate alignment by providing semantically meaningful reward shaping early in training. While this effect requires further investigation, it suggests that inference-based rewards can, in some cases, bootstrap learning more effectively than raw task signals. The corresponding learning curves are illustrated in Figure 7, highlighting the faster rise in episode length under inferred rewards.

**Additional Results.** To further probe the flexibility of InfeRL, we examine its ability to induce novel and compositional behaviors (see Appendix C.1 and C.2 for details). In the MuJoCo Ant domain, natural language prompts such as "spin in place while staying balanced" lead to qualitatively new locomotion strategies that differ from the default forward gait, while in Walker2D, vague instructions like "walk upright steadily" produce plausible but unintended backward walking. These case studies highlight both the promise of natural language rewards in guiding complex behaviors and the risks posed by underspecified prompts.

We also evaluate multi-objective and compositional instructions in modified CartPole environments. In FireWater, agents must balance the pole while positioning near fire and away from water, whereas in MultiCue they must additionally consider umbrellas and rain. PPO agents show partial to strong alignment with these goals, with success depending on the salience of visual cues, while DQN reliably solves the simpler base CartPole task. Together, these findings demonstrate that InfeRL generalizes beyond standard control to instruction-driven, multi-objective scenarios, but also underscore the importance of prompt clarity and cue distinctiveness.

## C.5 SUMMARY OF FINDINGS

Our experiments yield several insights into the effectiveness and limitations of inference-based reinforcement learning.

First, we find that monotonicity provides a useful diagnostic for evaluating the reliability of inferred rewards. In simple domains such as CartPole and InvertedPendulum, monotonicity scores approach unity (Kendall's $\tau > 0.92$, Spearman's $\rho > 0.98$), reflecting strong agreement between inferred and true returns. In more complex domains such as Ant and Walker2D, monotonicity remains informative: well-chosen prompts yield higher agreement with ground-truth returns, while ambiguous or poorly designed prompts break the property. This suggests that monotonicity analysis can guide the design and selection of prompts, offering a principled alternative to ad-hoc specification.

Second, the algorithmic comparison between PPO and DQN validates our theoretical predictions. PPO achieves stable learning whenever trajectory-level monotonicity is preserved, performing comparably under inferred and true rewards in CartPole and InvertedPendulum. In contrast, DQN exhibits slower convergence and reduced robustness in settings where the Markov property is violated, such as CartPole-FireWater and CartPole-MultiCue. These results demonstrate that policy gradient methods are better aligned with the properties of inferred rewards, while step-wise value-based methods remain fragile.

Third, our case studies illustrate both the promise and limitations of instruction-driven behaviors. In Ant, the agent successfully learns to rotate in place under a natural language prompt, demonstrating that novel, non-default behaviors can emerge without hand-engineered rewards. In Walker2D, however, ambiguous instructions result in backward walking, a behavior consistent with the inferred reward but misaligned with implicit expectations. These outcomes highlight both the flexibility of semantic rewards and the importance of addressing non-Markov dependencies and prompt ambiguity.

Taken together, these findings suggest that monotonicity provides a unifying principle for diagnosing reward reliability, that trajectory-based methods are particularly well-suited to inference-based settings, and that careful prompt design remains essential for realizing the full potential of natural language reward specification.

