# OpenReview forum: "Inference-based Rewards for Reinforcement Learning"
_ICLR.cc/2026/Conference — Submitted to ICLR 2026_

### Official Review · Reviewer_UKwQ · 2025-10-26

**Soundness:** 2
**Presentation:** 2
**Contribution:** 2
**Rating:** 2
**Confidence:** 3

**Summary:**

This paper studies inferred rewards framework to analyze the use of Vision-Language Models (VLMs) as reward providers. The paper proposes two key properties: monotonicity (two trajectories order based on true reward should remain the same based on the any provided reward mechanism) and the Markov property. The paper claims that policy gradient-based methods are more robust to the non-Markovian nature of VLM rewards because they depend on trajectories, rather than step-wise Bellman updates which are fully dependent on the Markovian nature. Experimental results support this claim.

**Strengths:**

1. The paper attempts to systematize the discussion around VLM-based rewards. By explicitly identifying and naming monotonicity and the non-Markovian nature of these rewards as key properties, it provides concepts to analyze this problem. While the necessity the non-Markovian in VLM have been noted in prior work [1], [2], the paper aims to frame this. Moreover, the monotonicity they defined seems to be obvious property we desired, which states that optimizing proposed reward function should imply 'true' reward function.

2. The paper aims to bridge its conceptual framework with empirical validation. It doesn't just discuss monotonicity in the abstract; it proposes concrete metrics (e.g., pairwise agreement, Kendall's $\tau$) for measuring it.

**Weaknesses:**

1. This paper tries to frame the necessary properties for VLM-based rewards, but there is no rigorous theoretical arguments on it. Also, some of claim can be more clear. For example, line 255, the equation shows: $R\left(\tau_1\right)>R\left(\tau_2\right) \Rightarrow  \hat{R}\left(\tau_1\right)>\hat{R}\left(\tau_2\right)$. I think they want to claim 'order preserving' regarding some rewards. Intuitively, the same logic, but different equation looks much clear in my opinion; $\hat{R}\left(\tau_1\right)\leq \hat{R}\left(\tau_2\right) \Rightarrow  R\left(\tau_1\right) \leq R\left(\tau_2\right)$. Since even if we change the original equation to $R\left(\tau_1\right) \leq R\left(\tau_2\right) \Rightarrow  \hat{R}\left(\tau_1\right) \leq \hat{R}\left(\tau_2\right)$. Their claim should still hold.

2. In section 4.3, they mention policy gradient methods and Q-learning based methods. However, during experiments, they work on PPO and DQN, where PPO depends on Actor-Critic method relying on value difference lemma, which is not pure policy gradients method as they rely on Bellman-update as well.

3. To measure monotonicity, they still need "True return of the given trajectory", which leads to impracticality when we don't have any true reward function.

**Questions:**

1. Your measurement method still need 'true' reward function, how much this will be useful when we don't have any access to the true reward function to solve the problems.

2. On line 255, conceptually, the following equation looks much easier to interpret even though they have the same meaning; $\hat{R}\left(\tau_1\right) \leq \hat{R}\left(\tau_2\right) \Rightarrow  R\left(\tau_1\right) \leq R\left(\tau_2\right)$, which simply says order preserving regarding reward. I think even if you change the original equation in Line 255 to $R\left(\tau_1\right) \leq R\left(\tau_2\right) \Rightarrow  \hat{R}\left(\tau_1\right) \leq \hat{R}\left(\tau_2\right)$, your definition of 'monotonicity' should remain the same. Is this correct?

[1] J. Beck, "Offline RLAIF: Piloting VLM Feedback for RL via SFO," arXiv preprint arXiv:2503.01062, 2025.

[2] H. Kang, E. Sachdeva, P. Gupta, S. Bae, and K. Lee, "GFlowVLM: Enhancing Multi-step Reasoning in Vision-Language Models with Generative Flow Networks," arXiv preprint arXiv:2503.06514, 2025.

---

> ### Author Response · Authors · 2025-12-02
> **Response to Reviewer UKwQ**
>
> **Summary of Revisions:**
> The manuscript has been revised to incorporate the reviewer’s suggestions regarding the theoretical justification for policy improvement under monotonicity, clarification of the monotonicity definition, distinctions between REINFORCE, PPO, and DQN, and the practical role of monotonicity analysis. Section 4.1 now adopts the reviewer’s order-preserving formulation. Section 4.2 includes a lemma connecting monotonicity to policy improvement for trajectory-level policy gradients. Section 4.3 clarifies the distinction between actor–critic and value-based methods, and new REINFORCE experiments in Section 5.3 empirically validate the theory.
>
> **Point 1: Theoretical Justification for Policy Improvement Under Monotonicity**
> The revised manuscript provides a formal statement establishing when optimizing an inferred reward leads to improvement under the true reward. Using the order-preserving condition suggested by the reviewer,  $\hat{R}(\tau_1) \le \hat{R}(\tau_2) \Rightarrow R(\tau_1) \le R(\tau_2)$,  Section 4.2 presents a lemma showing that any update increasing the expected inferred return also increases the expected true return. The proof sketch explains that trajectory-level policy gradient methods increase the probability of trajectories with higher inferred return, and monotonicity ensures that these trajectories also have higher true return. The analysis does not rely on Markovian rewards and applies directly to REINFORCE. PPO retains this alignment because its critic serves as a variance-reduction baseline rather than a Bellman target. New REINFORCE experiments in Section 5.3 verify this prediction empirically.
>
> **Point 2: Clarification of the Monotonicity Equation**
> The revised manuscript adopts the reviewer’s clearer inequality form  $\hat{R}(\tau_1) \le \hat{R}(\tau_2) \Rightarrow R(\tau_1) \le R(\tau_2)$. This presentation removes unnecessary asymmetry and emphasizes that the key requirement is preservation of the trajectory ordering induced by the true reward. These clarifications appear in Section 4.1.
>
> **Point 3: Distinguishing REINFORCE, PPO, and DQN**
> Section 4.3 now clearly distinguishes pure policy gradient methods, actor–critic methods, and value-based methods. The theoretical discussion refers specifically to REINFORCE, which uses unbiased Monte Carlo returns and makes no Bellman assumptions. Under the monotonicity condition, REINFORCE is the exact algorithmic counterpart of the theoretical analysis.
>
> PPO occupies a middle ground. Although it uses a value function, the critic acts only as a baseline and does not impose Bellman-consistent targets. As a result, PPO inherits the robustness properties of trajectory-level policy gradients and does not require the reward to be Markovian. This explains its stability in our experiments.
>
> In contrast, DQN relies on Bellman-consistent per-step rewards, which are not available under window-based VLM-inferred rewards. This leads to inconsistent targets and the instabilities observed in Section 5.4. To make the distinction concrete, we added new REINFORCE experiments in Section 5.3, which match PPO’s behavior and validate the theory.
>
> **Point 4: Practical Value of Monotonicity Analysis**
> Monotonicity is not required during training or deployment and does not assume access to true reward in real applications. Its role is as an offline diagnostic tool in benchmark environments where true reward is available. VLM-based rewards vary with prompt phrasing and context, and some prompts yield aligned rewards while others produce misleading or degenerate behavior. Prior work identifies these issues only after full RL training.
>
> Monotonicity offers a quantitative diagnostic for whether an inferred reward preserves the intended ordering of trajectories before training begins. Even when true rewards are unavailable at deployment, monotonicity remains valuable during development for prompt selection, model comparison, and reward debugging.

---

### Official Review · Reviewer_Jgo4 · 2025-10-28

**Soundness:** 3
**Presentation:** 3
**Contribution:** 3
**Rating:** 6
**Confidence:** 3

**Summary:**

This paper addresses the common misalignment between the reward function and the true intended target in reinforcement learning (RL). The authors propose an algorithm that replaces the original reward with a VLM-based reward, which evaluates the alignment between a trajectory window and a text-specified goal. The paper also discusses the conditions that ensure the effectiveness of such rewards and analyzes which RL algorithms are suitable for this inferred reward through theoretical reasoning and experiments, with an emphasis on the monotonicity of the inferred reward and the Markovian requirement.

**Strengths:**

1. The paper provides a practical and simple approach for leveraging the prior knowledge of VLMs to generate high-quality reward signals.

2. Theoretical analysis of the proposed method is provided, explaining why and when it achieves better performance. In addition, the paper offers practical methods for quantifying the monotonicity of the inferred reward.

3. The experimental design is detailed and effectively supports the analysis and assumptions.

**Weaknesses:**

1. Although the experiments provide supportive and intuitive results for the analysis (monotonicity and Markovian condition), there are few experiments that directly compare performance under the proposed reward with that under traditional rewards. More experimental results are expected to further confirm the effectiveness of the framework.

2. The experimental results show reduced correlation in more complex environments. Does this imply that the proposed method is currently limited to simpler tasks?

3. The quality of the inferred reward largely depends on the pretrained CLIP model, which is not fine-tuned during the InfeRL process. If the VLM model becomes the bottleneck, how can performance be further improved?

**Questions:**

See the weaknesses above.

---

> ### Author Response · Authors · 2025-12-02
> **Response to Reviewer Jgo4**
>
> **Summary of Revisions:**
> The manuscript has been updated to address the reviewer’s concerns regarding direct comparison to traditional rewards, reduced correlation in complex environments, and dependence on the underlying VLM. New experiments in Sections 5.2 and 5.3 compare PPO performance under inferred and environment rewards across all benchmark tasks, together with corresponding monotonicity scores. Section 5.4 and 5.5 add an analysis of prompt sensitivity and temporal window size, clarifying how monotonicity and quasi-Markov structure vary with visual complexity.
>
> **Point 1: Direct Comparison with Traditional Rewards**
> The revised manuscript includes direct comparisons between PPO trained with inferred rewards and PPO trained with the environment reward across the benchmark tasks (numerical comparisons in Table 2). These results show that when the inferred reward satisfies the monotonicity and quasi-Markov conditions identified in this work, PPO achieves performance comparable to training with the environment reward. The corresponding monotonicity scores reported in Section 5.2 complement these comparisons and help explain when inferred rewards support effective learning.
>
> **Point 2: Reduced Correlation in Complex Environments**
> Lower monotonicity scores in Walker2D and Ant are consistent with the purpose of InfeRL, which is to characterize when inferred rewards are suitable for reinforcement learning. These environments include visually complex states and multi-factor dynamics that make it more difficult for a pretrained VLM and prompt to produce consistent trajectory rankings. Section 5.4 adds experiments that vary prompt specificity and the temporal window size $k$, showing that both adjustments increase monotonicity and, in turn, improve learning performance. These findings confirm that reduced correlation results from reward quality rather than limitations of the InfeRL framework. Additional observations are provided in Appendix C.
>
> **Point 3: Dependence on the Pretrained VLM**
> The revised manuscript clarifies that InfeRL is agnostic to the specific VLM used for reward inference. The experiments use zero-shot CLIP to remain consistent with prior VLM-as-reward studies. These considerations are orthogonal to the contribution of InfeRL, which is to identify conditions under which any inferred reward is suitable for reinforcement learning. More capable VLMs are expected to satisfy the monotonicity and quasi-Markov conditions more often, but the framework itself remains valid regardless of the chosen model.

---

### Official Review · Reviewer_dhRr · 2025-10-29

**Soundness:** 2
**Presentation:** 3
**Contribution:** 2
**Rating:** 6
**Confidence:** 3

**Summary:**

The paper proposes Inference-Based Reinforcement Learning (InfeRL): instead of using environment rewards, an agent infers rewards from a pretrained vision–language model (VLM) conditioned on a natural-language goal. The core theoretical stance is that two structural properties largely determine whether such rewards are useful for learning: (i) trajectory-level monotonicity—the inferred return preserves the ordering of trajectories by true performance—and (ii) (quasi-)Markovianness—per-step reward can be made Markovian with a bounded temporal window. The paper argues that policy-gradient methods (e.g., PPO) depend mainly on trajectory-level signals and thus tolerate non-Markov rewards better than value-based methods (e.g., DQN), whose Bellman backups assume Markovian per-step rewards. Empirically, the authors compute rank-correlation metrics between true and inferred returns and compare PPO vs. DQN on CartPole variants and MuJoCo control.

**Strengths:**

- Recasting reward design as reward inference aligns with a fast-growing line of work that uses pretrained models or language to specify goals. The paper’s explicit articulation of trajectory-level monotonicity as a practical diagnostic is helpful for practitioners who currently rely on ad‑hoc prompt tinkering.
- The PPO vs. DQN contrast is plausible: policy gradients can optimize any scalar trajectory return (REINFORCE/GPOMDP), whereas Bellman targets fail when per-step rewards are history-dependent.
- Using Kendall’s τ / Spearman’s ρ to validate whether a prompt yields a usable reward signal is straightforward and reproducible.

**Weaknesses:**

- Much of the paper’s core message—“optimize trajectory-level signals from learned or inferred rewards; PG tolerates non-Markov structure better than Q-learning”—has been made in previous works.
  - Preference-based RL / RLHF already builds on trajectory orderings (pairwise comparisons) and uses policy-gradients over those signals [1].
  - VLM‑as‑reward and VLM‑feedback papers (zero‑shot rewards, RoboCLIP, RL‑VLM‑F) already examine robustness/instabilities and how to elicit better signals; comparisons here are too light [2-4].
- The paper motivates monotonicity conceptually but does not establish conditions under which monotonicity of returns implies true policy improvement under stochastic sampling (e.g., how much anti‑monotone noise can PPO tolerate?). A tighter connection to classical reward transformation invariance would help distinguish what is genuinely new (monotone trajectory ranking) vs. known (potential‑based shaping, affine transforms).
- The Markov critique of DQN hinges on the reward’s history dependence, but state augmentation (frame stacking / recurrent Q‑nets) and reward delay often restore Bellman validity. The experiments don’t report whether DQN received comparable temporal context or whether reward delays were tuned, so the conclusion may partly reflect architecture mismatch, not an inherent limitation.

# References

[1] Deep reinforcement learning from human preferences.

[2] RL-VLM-F: Reinforcement Learning from Vision Language Foundation Model Feedback.

[3] RoboCLIP: One Demonstration is Enough to Learn Robot Policies.

[4] Vision-Language Models are Zero-Shot Reward Models for Reinforcement Learning.

**Questions:**

Please refer to the weaknesses.

---

> ### Author Response · Authors · 2025-12-02
> **Response to Reviewer dhRr**
>
> **Summary of Revisions:**
> In response to the reviewer’s comments, we expanded Section 2 (Related Work) to clarify how InfeRL differs from preference-based RL, RLHF, and existing VLM-as-reward approaches by introducing explicit reward properties that determine when inferred rewards are suitable for reinforcement learning. Sections 4.1 and 4.2 now include the reviewer-suggested order-preserving formulation of monotonicity and a lemma establishing the conditions under which policy gradient updates improve the true return, addressing the original theoretical gap. To ensure fairness when comparing value-based and policy-gradient methods, Section 5.4 and Appendix A describe matched frame-stacked inputs for DQN and PPO and include new experiments varying the temporal window size $k$, confirming that DQN’s sensitivity arises from quasi-Markov rewards rather than architectural choices.
>
>
> **Point 1: Relation to Prior Work**
>
> The revised manuscript clarifies how InfeRL differs from preference-based RL, RLHF, and VLM-as-reward approaches. Prior work focuses on constructing or improving reward models through preference learning, VLM scoring, or language feedback, but does not provide conditions under which such inferred rewards are suitable for reinforcement learning.
>
> InfeRL introduces two explicit reward properties, order-preserving monotonicity and quasi-Markov structure, and shows how these properties govern algorithmic stability for trajectory-based and value-based methods. Section 2 situates InfeRL relative to prior work, and Sections 4.1 to 4.3 develop the theory and diagnostics, including Kendall’s $\tau$ and Spearman’s $\rho$, that predict reward viability prior to RL training. This diagnostic and theoretical framing is not addressed in prior VLM-reward or preference-RLHF methods.
>
>
> **Point 2: Theoretical Grounding of Monotonicity and Policy Improvement**
>
> The revised manuscript includes a formal condition linking monotonicity to policy improvement. Section 4.2 adopts the order-preserving definition of monotonicity and provides a lemma showing that  $\hat{R}(\tau_1) \le \hat{R}(\tau_2) \Rightarrow R(\tau_1) \le R(\tau_2)$  implies that any update increasing the expected inferred return also increases the expected true return.
>
> The proof sketch clarifies that trajectory-level policy gradient methods remain aligned with the true task objective as long as monotonicity violations do not invert the ranking of high-value trajectories. This addition directly addresses the reviewer’s request and delineates the robustness limits of policy gradients under inferred rewards.
>
>
> **Point 3: Fairness of the DQN Baseline and the Markov Property**
>
> The revised manuscript ensures architectural fairness and directly addresses the reviewer’s concerns regarding the Markov property. Both DQN and PPO receive identical frame-stacked observations with the same temporal window size $k$. Section 5.4 evaluates performance as $k$ varies, and DQN performance improves with additional temporal context, consistent with the reviewer’s intuition.
>
> However, Section 4.3 clarifies that the inferred reward is only quasi-Markov and may violate Bellman consistency even with augmented states. PPO and REINFORCE, which optimize full trajectory returns, remain stable across window sizes as shown in Figure 1, while DQN remains sensitive due to its dependence on per-step targets. This demonstrates that the observed differences stem from algorithmic sensitivity to Markovity rather than architectural mismatch.
>
>
> **Point 4: Distinction from Preference-Based RL and RLHF**
>
> The revised manuscript clarifies that preference-based RL and RLHF focus on learning a reward model from comparisons but do not analyze when such inferred rewards are suitable for reinforcement learning. Section 2 now highlights this distinction.
>
> InfeRL introduces two reward properties, order-preserving monotonicity and quasi-Markov structure, which characterize when an inferred reward can support stable optimization. Section 4.1 explains how monotonicity metrics serve as diagnostic tools rather than training objectives. This type of analysis is absent from preference learning and RLHF, which concentrate on reward construction rather than reward suitability.

---

> > ### Author Response · Authors · 2025-12-02
> > **Response to Reviewer dhRr - Part 2**
> >
> > **Point 5: Distinction from RoboCLIP, RL-VLM-F, and VLM-Reward Robustness Studies**
> >
> > Prior VLM-as-reward works such as RoboCLIP and RL-VLM-F investigate how to construct VLM-derived rewards and document empirical failure cases, but they do not provide general reward properties that explain when such signals support stable RL. Section 2 clarifies this distinction.
> >
> > InfeRL identifies two fundamental properties, monotonicity and quasi-Markovity, that characterize the suitability of inferred rewards across algorithms. Sections 4.1 to 4.4 show how these properties explain behaviors observed in prior VLM-reward studies, including the robustness of trajectory-based methods and the fragility of value-based ones. In this way, InfeRL provides a unified framework that explains why certain inferred rewards succeed, when they fail, and which RL algorithms can make effective use of them.

---

### Official Review · Reviewer_NjXL · 2025-10-29

**Soundness:** 2
**Presentation:** 3
**Contribution:** 3
**Rating:** 6
**Confidence:** 4

**Summary:**

This paper introduces a framework, InfeRL (Inference-based Reinforcement Learning), that characterizes the conditions under which VLM-based rewards, particularly similarity-based rewards, are effective for policy learning. The authors identify two key factors for stabilizing policy learning: the monotonicity of trajectory-level VLM-based returns and the Markov property of per-step VLM-based rewards. Based on these conditions, the paper examines which classes of RL algorithms, including policy gradient and Q-learning, are most suitable for training with such rewards. To quantify reward monotonicity, the authors introduce metrics such as pairwise agreement and rank correlation (Kendall's $\tau$ and Spearman's $\rho$). The framework is evaluated on two classical control tasks (CartPole and InvertedPendulum) and two locomotion tasks (Walker2D and Ant). Results indicate that policy gradient methods are generally more robust and effective for learning with VLM-based rewards.

**Strengths:**

1. The paper is clearly written and easy to follow, and the problem is well-motivated.
2. It addresses an important question: under what conditions are VLM-based rewards useful for reinforcement learning. From an algorithmic perspective, the paper provides insightful analyses of why common RL algorithms succeed or fail when trained with VLM-based rewards.
3. The introduced metrics for measuring monotonicity are valuable, as they can help design better language prompts or select appropriate VLMs.
4. The work has the potential to make a meaningful contribution, though it would benefit from further clarification of the framework and experimental setup.

**Weaknesses:**

1. The proposed framework uses the CLIP model as a VLM, but since the CLIP model takes a single image as input, it is unclear how it is used to derive rewards for a trajectory fragment. Is the reward obtained by averaging the individual similarity scores across frames?
2. The paper mentions goal-conditioned policies within the InfeRL framework and their ability to generalize at test time, but it appears that this type of policy was not actually used in the experiments.
3. In the experiment investigating monotonicity (Section 5.2), the evaluated trajectories are generated by a random policy. Is this sufficient to include both successful and failed trajectories? For instance, in Walker2D, random actions are likely to cause the robot to collapse. Therefore, the statement that "the inferred rewards distinguish between successful walking and collapsed robots" seems inappropriate. Could the authors elaborate on how this benchmark dataset was constructed?
4. Although Table 1 reports monotonicity scores across different environments, there is no analysis of how the degree of monotonicity correlates with the agent's actual performance. Including such insights would greatly strengthen the paper. For example, how does varying the degree of monotonicity (e.g., by adjusting language prompts or selecting different VLM models) affect the overall performance of the agent?
5. There is limited investigation into the Markov property. If the MDP is augmented to make it more Markovian (e.g., by using the same history or future context for both the reward and Q-function, as mentioned in Section 4.2), would the stability of DQN improve?
6. The experiment in Walker2D (Section 5.4) is not particularly compelling in supporting the paper's contribution, primarily due to the lack of ground-truth rewards. Without these, it is unclear how the monotonicity metrics are being utilized in this context. Additionally, the learning curve referenced in Line 456 seems to be missing from Figure 2.
7. Missing implementation details: What is the window size value? How does the VLM provide rewards for a trajectory, at the end (sparse) or at each timestep (dense)? Without these details, it may be difficult to understand the Markov property of VLM-based rewards.

Minor comments:
- In line 306, should $s_{t+k}$ be $s_{t+k-1}$?
- In Table 2, is the Cartpole-Base result missing?

Suggestion:
- The experimental design could be strengthened: The paper focuses on two axes of VLM-based rewards: monotonicity and Markov properties. If we consider each property as being either "strong" or "weak", the current experiments seem to focus primarily on strong monotonicity and weak Markov property. To provide deeper insights, it would be helpful if the experiments could also explore cases where both properties vary, reflecting a broader range of scenarios.

**Questions:**

Please see the Weakness.

---

> ### Author Response · Authors · 2025-12-02
> **Response to Reviewer NjXL**
>
> **Summary of Revisions:**
> The manuscript has been revised to address the reviewer’s requests for clearer descriptions of reward construction, goal usage, monotonicity evaluation, the relationship between monotonicity and downstream learning, the role of quasi-Markov structure, and the interpretation of the Walker2D example. Section 3 now provides explicit details on frame stacking and reward computation. Sections 5.2 to 5.4 include additional experiments that vary prompt monotonicity and temporal window size. Section 6 clarifies how the framework extends to stronger VLMs. Appendix A and Appendix C consolidate implementation details, trajectory dataset construction, and combined analyses of monotonicity and Markov structure.
>
> **Point 1: How CLIP Produces Rewards from Trajectory Fragments**
>
> The revised manuscript clarifies the reward construction mechanism. At each timestep, a trajectory fragment of length $k$ is converted into a CLIP input by stacking the $k$ frames along the channel dimension to form a single multi-channel tensor. This tensor is passed once through the CLIP vision encoder to compute a dense inferred reward. Section 3 now describes this operation explicitly, and Appendix A includes a schematic illustrating the stacking mechanism. Section 4.2 also notes that this construction yields a $k$-quasi-Markov reward, linking the procedure to the Markov property discussion.
>
> **Point 2: Clarification of “Goal-Conditioned Policies’’**
>
> The manuscript clarifies that the natural language goal influences the agent only through the inferred reward $f_{\text{inf}}(\tau_t, g)$. The policy network does not receive the goal as an input and is therefore not a goal-conditioned policy in the typical sense. All references to “goal-conditioned policy’’ have been replaced with the more precise phrase “policy optimized using a goal-specified inferred reward’’ in Sections 3 and 5.
>
> **Point 3: Monotonicity Evaluation Dataset**
>
> The revised manuscript explains how trajectories are collected for monotonicity evaluation. While random-policy rollouts provide adequate behavioral diversity in simple control benchmarks, they are limited in locomotion tasks. The updated monotonicity dataset therefore combines random trajectories, rollouts from PPO checkpoints at multiple training stages, and episodes from partially trained agents that occasionally achieve upright locomotion. Section 5.2 describes this dataset construction, and the updated monotonicity scores are reported in Table 1.
>
> **Point 4: Relationship Between Monotonicity and RL Performance**
>
> The manuscript now includes explicit analysis connecting monotonicity to downstream RL performance. Table 3 and Section 5.5 show that final PPO performance increases with monotonicity across evaluated prompts and environments, consistent with the trajectory-level alignment implied by the monotonicity condition. DQN performance improves only when both monotonicity and approximate Markov structure are present. These results clarify how reward properties influence learning dynamics.
>
> **Point 5: Investigation of the Markov Property**
>
> Additional experiments in Section 5.4 vary the temporal window size $k$ to investigate quasi-Markov structure. Increasing $k$ incorporates more recent dynamics into the reward computation and improves DQN performance, consistent with the dependence of value-based methods on Bellman-consistent per-step rewards. PPO and REINFORCE remain stable across all values of $k$, reflecting their trajectory-level handling of rewards. Section 4.2 expands the discussion of how temporal windows induce quasi-Markov structure.
>
> **Point 6: Walker2D Experiment and Interpretation of Results**
>
> The Walker2D example is presented as an auxiliary illustration of how ambiguous language prompts can lead to qualitatively valid yet unintended behaviors under inference-based rewards. Since the primary empirical evaluation is supported by expanded monotonicity analysis, Markov structure experiments, and comparisons across PPO, REINFORCE, and DQN, the Walker2D results have been moved to Appendix B. The revised text clarifies the use of episode length as a standard stability proxy.

---

> > ### Author Response · Authors · 2025-12-02
> > **Response to Reviewer NjXL - Part 2**
> >
> > **Point 7: Missing Implementation Details**
> >
> > The revised manuscript now includes a consolidated summary of implementation details in Appendix A. Section 3 and Appendix A specify the temporal window size $k$, the frame-stacking procedure, dense per-timestep reward computation, and how trajectory returns are accumulated. The interaction between window size and quasi-Markov structure is also clarified, ensuring reproducibility and addressing the reviewer’s concerns.
> >
> > **Point 8: Exploring Combinations of Monotonicity and Markov Structure**
> >
> > The manuscript incorporates experiments requested by the reviewer to explore combinations of strong and weak monotonicity with varying degrees of Markovity. We vary the prompt specification to induce different monotonicity levels and adjust the temporal window size $k$ to control quasi-Markov structure. Appendix C reports these combined experiments and shows that PPO and REINFORCE remain robust across a wide range of monotonicity values, while DQN is sensitive to both monotonicity and Markovity. Sections 5.2 and 5.4 summarize these findings in the main text.

---

### Author Response · Authors · 2025-12-02
**Summary of Rebuttal and Revisions**

The revision addresses all concerns raised by the reviewers. We strengthened the theoretical grounding of InfeRL by adopting the order-preserving definition of monotonicity suggested by the reviewers and adding a lemma that formally links monotonicity to policy improvement under trajectory-level policy gradients (**dhRr**, **UKwQ**). We clarified the roles of REINFORCE, PPO, and DQN by distinguishing pure policy gradient methods from actor–critic and value-based approaches, and we added new REINFORCE experiments that empirically validate the theory (**dhRr**). The definition of monotonicity was rewritten as a ranking-preservation property, and Section 6 now explains the practical value of monotonicity as an offline diagnostic tool rather than a requirement for real-world training (**UKwQ**).

We expanded the explanation of reward construction and clarified how the goal affects the agent only through the inferred reward signal (**NjXL**). Section 3 now explicitly describes the frame-stacking procedure, and all references to “goal-conditioned policy’’ have been replaced with “policy optimized using a goal-specified inferred reward.” The monotonicity evaluation was strengthened by incorporating random rollouts, partially trained PPO checkpoints, and early-stage locomotion episodes to improve behavioral diversity (**NjXL**). Additional experiments vary both prompt phrasing and temporal window size to jointly explore monotonicity and Markov structure, as suggested by the reviewer (**Jgo4**). These results, summarized in Sections 5.2 and 5.4 and detailed in Appendix C, show that PPO and REINFORCE remain robust across a wide monotonicity range, while DQN is sensitive to both monotonicity and approximate Markovity. Direct comparisons between inferred and environment rewards were added in Sections 5.2 and 5.3, and secondary illustrations such as the Walker2D backward-walking behavior were moved to Appendix B (**Jgo4**).

All additions and modifications in the updated manuscript are marked in **blue** for ease of review. Individual point-by-point responses to each reviewer comment are provided below.

---

### Meta-Review · Area_Chair_MFPd · 2026-01-08

**Summary:**

- A major concern was the lack of formal proof connecting the proposed monotonicity condition to actual policy improvement. Reviewers argued that simply observing a correlation between rankings was insufficient without establishing that an update to increase the inferred return would reliably increase the true return. The authors added a lemma in Section 4.2 formally linking order-preserving monotonicity to policy improvement under trajectory-level policy gradients.

- Reviewers questioned the finding that DQN was more fragile than policy gradient methods. Specifically, there was a concern that DQN’s failure was due to an architectural disadvantage such as not receiving the same temporal context (frame stacking) as the VLM. The authors revised the experiments to ensure matched frame-stacked inputs for all algorithms, confirming that DQN's sensitivity arises from the quasi-Markov nature of the rewards rather than architecture.

- To measure monotonicity via pairwise agreement or rank correlation, the agent still requires access to the true gt return. This led to questions about the framework’s utility in real world settings where a true reward is not available. The authors clarified that monotonicity serves as an offline diagnostic tool for benchmark environments to guide prompt selection and reward debugging prior to deployment, rather than a runtime requirement.

- The claim that policy gradients tolerate non Markovian signals better than value based methods was noted as being well established in RL literature.

- Reviewers noted that monotonicity scores dropped significantly in complex locomotion tasks like Ant and Walker2D. This raised concerns about whether the framework's success is limited to simple control tasks with clear visual cues. The environmental and task diversity is quite limited.

**Reviewer Concerns:**

Primary one's that were addressed:
- Reviewers dhRr and UKwQ noted a lack of rigorous theoretical justification for monotonicity. The authors addressed this by adding a formal lemma proving that under the order preserving condition, policy gradient updates to the inferred return results in policy improvement.
- Reviewer dhRr suspected that DQN’s failure was due to a lack of temporal context. The authors resolved this by providing both DQN and PPO with identical frame stacked observations.

Outstanding:
- UKwQ argued that the framework remains impractical because measuring monotonicity still requires access to the true reward. While the authors framed this as an offline diagnostic tool, the concern remains about its easy and general-purpose applicability.
- Jgo4 noted that the zero shot nature of the VLM acts as a fixed bottleneck. This paper does not propose a method to fine-tune or improve the VLM itself if the initial semantic grounding is poor. This is a major limitation and no discussion / consideration is done.
- dhRr maintained that the insights noted about policy gradient vs Q-learning based method is well known in the RL literature, despite this setting being different due to VLMs. So the conceptual insights were thought of as incremental in nature.

**Reviewer Scores:**

- NjXL: Most questions resolved.
- dhRr: While the authors provided the requested theoretical Lemma in section 4.2 and ensured architectural fairness in the DQN/PPO comparison, this reviewer remained skeptical of the paper's core novelty.
- Jgo4: This reviewer specifically requested direct comparisons to traditional rewards and an analysis of how monotonicity affects complex tasks. This was largely addressed.
- UKwQ: Still had questions around implausibility of access to true rewards and deemed the core insights incremental in nature.

---

### Decision · Program_Chairs · 2026-01-26

Reject